# AGENTPACK: A DATASET OF CODE CHANGES, CO-AUTHORED BY AGENTS AND HUMANS

## ABSTRACT

Fine-tuning large language models for code editing has typically relied on mining commits and pull requests. The working hypothesis has been that commit messages describe human intent in natural language, and patches to code describe the changes that implement that intent. However, much of the previously collected data is noisy: commit messages are terse, human-written commits commingle several unrelated edits, and many commits come from simple, rule-based bots.

The recent adoption of software engineering agents changes this landscape. Code changes *co-authored* by humans and agents are often accompanied by substantially more explicit natural-language descriptions of intent and rationale. Moreover, when these changes land in public repositories, they are implicitly filtered by humans: maintainers discard low-quality commits to their projects.

We present AGENTPACK, a corpus of 1.3M code edits co-authored by Claude Code, OpenAI Codex, and Cursor Agent across public GitHub projects up to mid-August 2025. We describe the identification and curation pipeline, quantify adoption trends of these agents, and analyze the structural properties of the edits. Finally, we show that models fine-tuned on AGENTPACK can outperform models trained on prior human-only commit corpora, highlighting the potential of using public data from software engineering agents to train future code-editing models.

## 1 INTRODUCTION

We are interested in training large language models on code editing tasks, where the model is prompted with code and instructions on how to update the code. Prior work has shown that language models pretrained on code can be endowed with code-editing capabilities by fine-tuning them on code change data such as commits and pull requests mined from GitHub (Muennighoff et al., 2023; Cassano et al., 2024b). In fact, there is a significant volume of code change data available. CommitPack has 4TB of commits from GitHub, even though it limits itself to commits that only edit a single file (Muennighoff et al., 2023).

One may expect that code changes, which couple code with natural language descriptions of that change, capture human intent and understanding more deeply than just the final code in a repository. Unfortunately, this is not the case for the vast majority of code changes. Many programmers write very cryptic natural language descriptions, and commits authored by bots (before LLM agents) tend to update configuration files and not code (Dey et al., 2020). Therefore, although we can build datasets with terabytes of code changes, only a few gigabytes of these datasets have been deemed as useful training data (Muennighoff et al., 2023).

However, the landscape of coding activity has started to change. Claude Code was released in February 2025, followed shortly thereafter by several other coding agents, that are rapidly gaining popularity (Figure 1). Their emergence has introduced a qualitatively different source of code change data: edits co-authored by agents and humans, with natural language descriptions that are often much more detailed than commit messages that humans write by themselves. In this paper we present AGENTPACK, a dataset of commits and pull requests authored by Claude Code, OpenAI Codex, and Cursor Agent across public GitHub repositories. Although these agents have only been available for a few months, they have already co-authored at least 60GB of code that we can identify that has been merged into open-source projects.

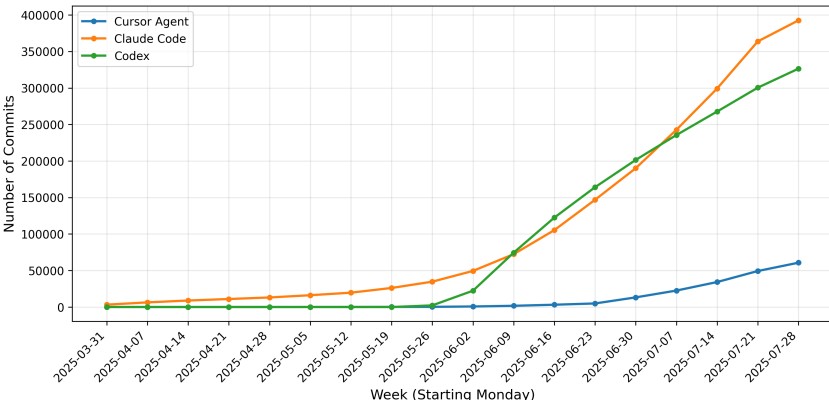

Figure 1: Total commits and new pull requests made by Cursor Agent, Claude Code, and Codex from launch to the indicated date.

The code changes in AGENTPACK are interesting for several reasons. 1) They are co-authored by agents and humans working together. Unlike synthetic data generation pipelines (§2), these changes are the outcomes of human-agent interactions, accepted by programmers and integrated into the codebase of hundreds of thousands of repositories. It is also likely that many changes have been further refined by humans. 2) Unlike code generated by an LLM-powered autocomplete, agent-written code is often accompanied with agent-written tests. The extent of testing is project dependent, but it is well understood that projects with good test infrastructure can help provide a strong external reward signal to an LLM agent (Shinn et al., 2023; Pan et al., 2025). 3) Finally agents output detailed natural language descriptions of their code changes. These descriptions often convey intent in more detail than descriptions written solely by humans. In sum, these three characteristics make AGENTPACK a diverse and high-quality dataset for training models on code editing tasks.

**Contributions**   In summary, we make the following contributions. (1) We present AGENTPACK, the first dataset of code changes co-authored by agents and humans, comprising 1.3M code edits from Claude Code, OpenAI Codex, and Cursor Agent across public GitHub repositories from April to mid-August 2025. (2) We develop a systematic pipeline for identifying, collecting, and curating agent-authored code changes from GitHub's public timeline (§3). (3) We provide a comprehensive analysis of AGENTPACK, quantifying the rapid adoption of software engineering agents in open-source development and characterizing their usage patterns, including the structural properties of code edits (§4.1), the distribution of file types and programming languages (§4.1), and the range of tasks being performed (§4.2). (4) Finally, We demonstrate the quality of agent–human collaborative data with fine-tuning experiments on AGENTPACK, showing significant improvements in code editing performance across models of varying sizes (§5).

## 2   BACKGROUND AND RELATED WORK

**Datasets for Fine-Tuning Models on Coding Tasks**   There are several datasets for fine-tuning models on coding tasks where the code, and often the prompt, is generated by an LLM (Luo et al., 2023; Wei et al., 2024; 2025a; Cassano et al., 2024a; Ahmad et al., 2025). What sets AGENTPACK apart from this prior work is its scale and diversity. With 1.3M code changes, it is nearly 2x larger than the most recent code dataset that is distilled from DeepSeek-R1 (Ahmad et al., 2025). Each training item in the aforementioned datasets is typically a single function or a solution to a competitive programming problem. In contrast, in AGENTPACK, each item is a code change comprising a few hundred lines that typically spans multiple files. With one exception (Cassano et al., 2024a), these datasets focus on Python programming tasks. Although AGENTPACK is dominated by Python and JavaScript, it also contains a substantial volume of code changes in low-resource programming languages.

**Reinforcement Learning for Coding Tasks**   Several recent papers apply reinforcement learning to programming problems (Boruch-Gruszecki et al., 2025; Wei et al., 2025b; Zeng et al., 2025; Gehring et al., 2024; Jain et al., 2025; Pan et al., 2025). Although this paper does not explore reinforcement learning, it may be possible to use AGENTPACK to train new software engineering agents. After all, the changes in AGENTPACK were authored by existing agents.

**Training Models to Edit Code**   There has been prior work on training models on the code editing task. Muennighoff et al. (2023) build a dataset of 4TB of GitHub commits, up to the year 2016, that affect a single source file. They use several rule-based filters to build *CommitPackFT*, which has 2GB of higher quality data for fine-tuning. Cassano et al. (2024b) filter the dataset further and complement it with newer code commits to build a dataset that we call *CanItEdit*. AGENTPACK, which has 60GB of code changes, is an order of magnitude larger than these prior datasets. Moreover, other metrics such as length of the natural language description and size of the edit indicate that the code changes in AGENTPACK are also more sophisticated.

**Benchmarking Models on Code Editing Tasks**   Unlike code synthesis benchmarks such as HumanEval and MBPP (Chen et al., 2021; Austin et al., 2021), which evaluate a model's ability to generate code from natural-language specifications, code editing evaluates a model's ability to update existing code given a natural-language description of the desired change. Muennighoff et al. (2023) introduced HumanEvalFix, which targets bug fixing, while (Cassano et al., 2024b) introduce the CanItEdit benchmark, which spans a broader diversity of code-editing tasks (Cassano et al., 2024b). In this paper, we evaluate on both benchmarks.

**Analyzing Open Source Code at Scale**   Hindle et al. (2009) were the first to classify code changes using statistical machine learning techniques. GitHub has since rapidly grown in popularity, and a lot of empirical work on code has studied code on GitHub. Lopes et al. (2017) found that up to 70% of the code on GitHub is duplicated, that the rate of duplication varies by language, and that JavaScript has the most duplicated code. The main reason that JavaScript code is duplicated is because programmers accidentally commit the `node_modules` directory. We encounter this in AGENTPACK and correct for it as we build the dataset.

Although LLM-powered software engineering agents are relatively new, rule-based bots have been popular on GitHub for many years. Dey et al. (2020) build a dataset of activity by 451 different bots. They find that most bots update configuration files and not source code. Software engineering agents are different in that they can be instructed to make arbitrary changes to a repository, and we observe this in both the types of changes they make and the variety of programming languages in which they write code.

## 3 BUILDING AGENTPACK

We construct AGENTPACK in five steps. First, we fetch the archive of the GitHub public timeline, hosted by GH Archive (Grigorik, 2025). We download events from April 1 2025 to August 15 2025. This timespan begins one week after Claude Code became generally available, and we intend to continue growing our dataset. These events, which are 262GB gzipped, contain metadata about pushes, pull requests, and other GitHub activity, but do not contain actual source code, which we fetch in a later step described below.

Second, we identify the events that are likely to involve activity by software engineering agents as follows.

1. **Claude Code** creates commits and typically signs them with `Co-authored-by: Claude <noreply@anthropic.com>` in the commit message. (There are other variations of this message that we also search for.) For each commit contained in a push, the GH Archive metadata has the commit message, the repository name, and the commit hash, which is all we need to later fetch the diff introduced by the commit.

2. **OpenAI Codex** operates differently and produces pull requests with detailed descriptions. In every pull request, it includes a link to the original conversation that begins with `chatgpt.com/codex/tasks`. We scan GH Archive for events that open pull requests,

| AGENTPACK | Totals | | Medians per item | | | |
|---|---|---|---|---|---|---|
| Agent / Dataset | Items (#) | Size (GB) | Files (#) | Patch size (lines) | Hunks (#) | Commit Message Length (chars) |
| Claude Code | 854,946 | 20 | 2 | 75 | 1.5 | 394 |
| Codex | 372,006 | 16 | 3 | 56 | 1.5 | 294 |
| Cursor Agent | 110,060 | 23 | 2 | 124 | 1.3 | 117 |
| **Total** | **1,337,012** | **59** | **2** | **70** | **1.5** | **323** |
| *Prior datasets* | | | | | | |
| CommitPackFT | 702,062 | 2 | 1 | 4 | 1.0 | 43 |
| CanItEdit | 43,971 | 0.2 | 1 | 7 | 1.0 | 57 |

Table 1: Summary statistics for AGENTPACK by agent. We also compute the statistics for the earlier CommitPackFT and CanItEdit datasets. AGENTPACK is significantly larger by every metric.

> which gives us the metadata we require: the original pull request description from Codex, and the base and head commit hashes. When we later fetch the diff, we get the diff for the entire pull request, which may include future commits by others.

3. **Cursor (Background) Agent** is identifiable similarly to Claude Code. It creates commits authored by `Cursor Agent <cursoragent@cursor.com>`. The GH Archive metadata has the commit message, commit hash, and repository name.

A small random sample of commits per agent is manually inspected to verify that agent attributions are correct.

Third, we download the repositories that have agent activity identified above. For each repository, we perform a shallow bare clone starting from April 1st 2025. We encounter several failures, typically due to repositories being deleted or marked private, but we still fetch 1.5TB of repositories, which consist of compressed code files. To filter for quality, we then process the cloned repositories to find and isolate only the commits that have been merged into the primary branch (typically main or master). We use merge status as a pragmatic heuristic to exclude unvetted or abandoned changes, for the commits that are not yet merged are typically not yet vetted or may have already been deemed poor quality.

Fourth, we use the commit hashes from the second step to get the associated git patch from the downloaded repositories. In addition, we filter out changes to files in the `node_modules` directory in each patch. This directory is where JavaScript projects store their dependencies in source form, and many repositories commit it–often inadvertently–which leads to substantial code duplication (Lopes et al., 2017). More importantly, although an agent may commit changes under `node_modules`, that code originates from third-party packages rather than the agent. Including such changes would misattribute work to the agent, so we exclude them.

As a final step, we perform a straightforward merge of agent activity metadata from GH Archive with the code we download directly from GitHub to build AGENTPACK. In the next section, we take a look at the variety of activity recorded in AGENTPACK.

## 4 EXPLORING AGENTPACK

In this section we examine both the code and commit messages in AGENTPACK, compare them to other datasets of code commits (Muennighoff et al., 2023; Cassano et al., 2024b), and dive into a selection of commits.

### 4.1 THE CODE IN AGENTPACK

**Summary statistics** Figure 1 shows the counts of commits made by each agent over time, and Table 1 summarizes the basic characteristics of AGENTPACK code edits. We report the total size and number of items by agent, as well as the following medians: (1) the number of files patched; (2) the patch size, which is the total number of lines added and removed in the diff, excluding

|  | Claude Code | Codex | Cursor Agent |
|---|---|---|---|
| **File count (N)** | 4,920,114 | 2,357,498 | 2,452,519 |
| **Share of files (fraction of column total)** | | | |
| TypeScript | 0.24 | 0.14 | 0.13 |
| JavaScript | 0.05 | 0.13 | 0.25 |
| *Data Files* | 0.10 | 0.09 | 0.17 |
| Python | 0.12 | 0.17 | 0.03 |
| Markdown | 0.11 | 0.09 | 0.05 |
| *Config Files* | 0.06 | 0.03 | 0.03 |
| HTML | 0.02 | 0.06 | 0.03 |
| Go | 0.04 | 0.03 | 0.01 |
| PHP | 0.01 | 0.02 | 0.02 |
| Rust | 0.03 | 0.01 | 0.00 |
| Java | 0.01 | 0.02 | 0.01 |
| CSS | 0.01 | 0.02 | 0.01 |
| C# | 0.01 | 0.02 | 0.00 |
| C | 0.00 | 0.01 | 0.01 |
| Shell | 0.02 | 0.01 | 0.00 |
| Ruby | 0.01 | 0.00 | 0.01 |
| Dart | 0.01 | 0.01 | 0.00 |
| Swift | 0.01 | 0.01 | 0.00 |
| *Other* | 0.14 | 0.14 | 0.23 |

Table 2: File counts and composition by file type in AGENTPACK. Values are column-wise shares in decimal form (i.e. between 0 and 1). Columns may not sum to exactly 1.00 due to rounding.

context; (3) the average number of hunks, which is the number of disjoint regions edited, per file; and (4) the length (in characters) of the natural language description that accompanies the code edit. We compute the same statistics for the CommitPackFT and CanItEdit datasets from prior work.

The table reveals that typical usage patterns are consistent across all three agents: the median values show that most patches touch just 2-3 files regardless of agent, with patch sizes ranging from 56-124 lines. This suggests that as they are used, all agents produce similarly scoped changes. The Cursor Agent seems to produce slightly larger patches than the other two.

**Comparison to other code editing datasets**  It is interesting to compare AGENTPACK to earlier code editing datasets, which we do in the lower portion of Table 1. Both CommitPackFT and CanItEdit deliberately excluded all edits that touched more than one file. However, even after normalizing by the number of files edited, the code edits in AGENTPACK are more complex. We also see that agent-written descriptions are nearly 10x longer than the human-written descriptions from earlier datasets. CommitPackFT and CanItEdit used several filters to remove short, unhelpful commit messages, thus the average description would have been even shorter without those filters. In contrast, AGENTPACK does not filter commit messages in any way, apart from filtering them by agents' signatures.

**The programming languages in AGENTPACK**  Table 2 shows the composition of AGENTPACK by agent, focusing on the most frequently occurring programming languages. This table is based on a classification of file extensions. The category *Config Files* includes Dockerfiles, `.ini` files, etc., the category *Data Files* includes `.json` files, `.csv` files, etc., and the category *Other* includes other programming languages. (See appendix A.1 for more details on the classification.) Each programming language in the *Other* category constitutes less than 1% of the files in AGENTPACK.

The dataset is dominated by JavaScript, TypeScript, and Python, which is not surprising. It is surprising the most popular language varies by agent: TypeScript for Claude Code, Python for Codex, and JavaScript for Cursor Agent. Markdown is one of the top extensions, which shows that agents are frequently tasked with writing documentation. Finally, apart from JavaScript, other web technologies such as HTML, PHP, CSS, Dart are also well represented.

| Language | Claude Code | Codex | Cursor Agent |
|---|---|---|---|
| OCaml | 9,196 | 20 | 93 |
| Julia | 5,507 | 139 | 118 |
| Fortran | 4,522 | 239 | 53 |
| Scala | 3,993 | 902 | 156 |
| R | 3,377 | 3,728 | 498 |
| Haskell | 775 | 43 | 145 |
| D | 252 | 920 | 4,049 |
| Racket | 121 | 116 | 1 |

Table 3: File counts of low-resource programming languages in AGENTPACK.

Although they are a small portion of the dataset, there is a non-trivial amount of activity with several low-resource languages. Table 3 reports file counts for these languages by agent. The numbers indicate that OCaml, Julia, Fortran, and Scala programmers use Claude Code disproportionately more than the other two agents. However, Cursor Agent seems to be used more by D programmers.

## 4.2 THE TYPES OF TASKS IN AGENTPACK

The commit messages and pull request descriptions in AGENTPACK have concise, LLM-written descriptions of the code changes that they produce. Examining these descriptions, it is clear that agents are used to perform a wide variety of tasks. As a first step toward understanding the types of tasks that programmers solve with software engineering agents, we automatically classify a sample of items from AGENTPACK as follows.

We first design a list of labels that describe the nature of code changes. The labels we use are as follows.

1. *Bugfix*: a change that corrects an error or defect in existing functionality.

2. *Configuration or build*: a change to build scripts, dependencies, environment setup or project configuration.

3. *Documentation*: a change that adds, updates, or clarifies project documentation, comments, or usage instructions.

4. *New feature*: a change that introduces new functionality, capability, or user-visible behavior to the system.

5. *Performance improvement*: a change that optimizes resource use, speed or scalability

6. *Refactor or code style*: a change that restructures code or adjusts formatting to improve readability or maintainability without altering behavior.

7. *Tests added or updated*: a change that adds or modifies tests to improve coverage, correctness, or reliability.

8. *Other*: a change that does not clearly fit into one of the above.

Real-world changes do not neatly fall into a just one of these categories, thus we assign multiple labels to each item.

We solve this multiclass classification problem by prompting Qwen3-4B-Instruct-2507 (Yang et al., 2025) to assign labels to each natural language description. Using DSPy (Khattab et al., 2024; 2022), we optimize the prompt for Qwen 3 to align its behavior with the much larger Sonnet 4 (Anthropic, 2025) model, using Jaccard similarity as the metric. We perform optimization using 200 commits to achieve 80% accuracy, and then use the optimized prompt to label a random sample of 5,000 commits per agent.

We plot the distribution of commit labels per agent in Figure 2. Overall, *new feature* was the most frequent label across all three agents, with Codex showing the highest concentration in this category, followed by Cursor Agent and Claude Code. The usage patterns diverge for other labels: Claude Code is used disproportionately for *bug fixes*, while Codex produces *documentation* and *tests added*

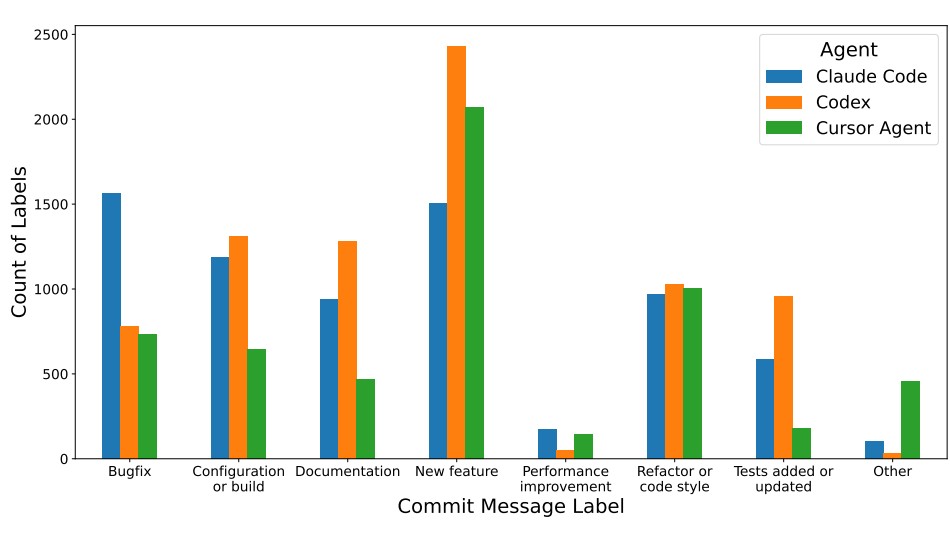

Figure 2: Distribution of commit labels across three coding agents (Claude Code, Codex, and Cursor Agent), based on a random sample of 5000 commits from each agents. Each bar represents the frequency of commits associated with a given label. A single commit may carry multiple labels, and counts reflect total label occurrences rather than single commits.

```
Fix Google Ads API customer status validation to handle numeric values

- Account status comes back as numeric value (2) instead of string 'ENABLED'
- Add support for both string and numeric status values
- Status codes: 0=UNKNOWN, 1=ENABLED, 2=SUSPENDED, 3=CLOSED
- Provide better error messages with human-readable status text
- This fixes the 'Account [REDACTED] is not enabled (status: 2)' error for
  campaign creation
```

(a) An example of a description written by Cursor that we classify as a *bugfix*, with personal identifiers redacted.

```
Complete Atlassian MCP integration and update default message

- Add Atlassian MCP server with Jira and Confluence search capabilities
- Integrate Atlassian MCP into AI service with token handling
- Update frontend default message to include Jira MCP search
- Fix TypeScript types and ESLint issues across all MCP servers
- Ensure symmetric OAuth cookie implementation across all providers

Generated with [Claude Code](https://claude.ai/code)
Co-Authored-By: Claude <noreply@anthropic.com>
```

(b) A description written by Claude Code that we classify with as *bugfix*, *new feature*, and *refactor or code style*.

```
## Summary
- instruct developers to run `npm install` before testing

## Testing
- `npm test`
```

(c) A description written by Codex that we classify as *documentation*.

Figure 3: Examples of three natural language descriptions authored by three different agents from AGENTPACK. In the captions above, we list the labels that we assigned to each of these descriptions for our classification.

*or updated* more frequently than the other two. Refactoring or code style changes were distributed nearly evenly across all three agents, whereas performance improvements are the rarest type of change.

Taken together, these usage patterns indicate that AGENTPACK is a rich, task-diverse resource spanning code maintenance (bug fixes, refactors, configuration), feature development, documentation, and testing, while still capturing rarer categories. For instance, we can expect that even for performance improvements there are approximately 35,000 changes in the entire dataset (if we extrapolate from the random sample of 15,000 code changes to a dataset of 1.3M). In the next section, we experiment with fine-tuning models on the Python subset of AGENTPACK. However, for future work it may be more interesting to focus on other programming languages or particular types of changes.

## 5 FINE-TUNING MODELS WITH AGENTPACK

We now consider the potential of training models on AGENTPACK. We focus on the code editing task, which is the task of producing updated code, conditioned on a prior version of the code, and a natural language instruction that describes the desired edit. We perform two set of fine-tuning experiments on AgentPack: one targeting Python and another targeting JavaScript. The Python experiment allows us to compare models fine-tuned on AGENTPACK to prior work on code editing by (Cassano et al., 2024b) and (Muennighoff et al., 2023). The JavaScript experiment allowing us to assess whether this potential generalize beyond Python.

**Dataset format** We fine-tune models on code edits that update at least one file in the target language (Python or JavaScript). We process each item as follows. We parse the Git patch for each item, to extract the list of updated files, the previous content and and the patched content. Note that a Git patch may not contain an entire file, but instead consist of several "hunks" surrounded by context. When this occurs, the content that we include has ellipses in between each hunk to indicate that there is unseen and unchanged code in the rest of the file. We prepare each training item as a prompt and completion, where the prompt has the natural language description, followed by the old contents, and the completion has the new contents. When an item patches several files, we precede each content with the file name. Finally, to manage the cost of training, we omit items that exceed 4,096 tokens. The final Python training set consists of 120 million tokens spanning 118,848 training items ($\approx 9\%$ of AGENTPACK); the JavaScript training set consists of 47 million tokens spanning 64000 training items ($\approx 5\%$ of AGENTPACK).

**Model selection, training hyperparameters, and evaluation settings** For Python training, to facilitate comparison with prior work, we train DeepSeekCoder Base (1.3B and 6.7B), which are the base models for EditCoder (Cassano et al., 2024b). In our replication, we get different scores ($\pm 0.02$) than those reported by Cassano et al. (2024b). From communication with the original authors, we conclude that our evaluation procedure is correct, thus we use our newly computed scores below. In addition, we train Qwen3-1.7B-Base and CodeLlama-7B-Base under the same evaluation protocol to provide a broader comparison with EditCoder and our dataset. For JavaScript training we used the same set of models without a prior work (EditCoder was only trained on Python).

We fine-tune all models with the AdamW optimizer, with learning rate $2 \times 10^{-5}$, global batch size $64$, a cosine learning rate schedule with warmup ratio $0.1$. For Python evaluations, DeepSeekCoder is trained for three epochs to remain consistent with prior work, while Qwen3 and CodeLlama are limited to 1000 training steps. For JavaScript, all models–Qwen3, CodeLlama, and DeepSeekCoder–are likewise trained for only 1000 steps to control computational cost. In all settings, we apply loss masking to the prompts.

For evaluation, we report pass@1 score computed using 20 completions sampled with temperature $0.2$ and top-$p$ sampling ($p = 0.95$). For the models trained to edit code, we evaluate zero-shot with a prompt template that matches the training format. For the base model, we provide a one-shot prompt with a single example (see Figure 5 in Appendix) when evaluating CanItEdit (see below).

**Benchmark Selection** We select two code editing benchmarks, HumanEvalFix and CanItEdit, as our evaluation set. HumanEvalFix (Muennighoff et al., 2023) is a variant of HumanEval (Chen et al., 2021), in which manually introduced bugs in the solutions serve as the problems, and the task is to

| Model | Training set | Benchmarks (Pass@1) | | |
|---|---|---|---|---|
| | | **HumanEvalFix-Py** | **CanItEdit** | **HumanEvalFix-JS** |
| DeepSeekCoder-1.3B-Base | N/A | 0.19 | 0.11 | 0.20 |
| | EditCoder | 0.20 | 0.29 | - |
| | AGENTPACK | **0.32** | **0.32** | **0.24** |
| DeepSeekCoder-6.7B-Base | N/A | 0.39 | 0.30 | 0.40 |
| | EditCoder | 0.45 | 0.42 | - |
| | AGENTPACK | **0.50** | **0.43** | **0.41** |
| Qwen3-1.7B-Base | N/A | **0.34** | 0.09 | 0.28 |
| | EditCoder | 0.33 | **0.10** | - |
| | AGENTPACK | **0.34** | 0.09 | **0.30** |
| CodeLlama-7B-Base | N/A | 0.24 | 0.21 | 0.25 |
| | EditCoder | 0.28 | 0.29 | - |
| | AGENTPACK | **0.36** | **0.30** | **0.31** |

Table 4: Pass@1 scores of models on HumanEvalFix-Py (Python), CanItEdit (Python) and HumanEvalFix-JS (JavaScript). Each block shows the base model without fine-tuning (N/A), with the EditCoder training set, and with our lightly filtered dataset (AGENTPACK).

| Model | Training format | HumanEvalFix-Py (Pass@1) |
|---|---|---|
| DeepSeekCoder-1.3B-Base | Ellipsis-hunk | **0.29** |
| | Aider-diff | 0.22 |
| DeepSeekCoder-6.7B-Base | Ellipsis-hunk | **0.52** |
| | Aider-diff | 0.36 |

Table 5: Pass@1 scores on HumanEvalFix-Py for DeepSeekCoder models fine-tuned with different training formats.

generate a correct, bug-free function. We use both Python and JavaScript subsets of HumanEvalFix. CanItEdit (Cassano et al., 2024b) is a Python-only evaluation dataset of 105 code editing problems spanning diverse domains in programming. We place the prompts used for each benchmark in Figures 4a and 4b, in the appendix.

**Results** Table 4 presents the results of fine-tuning models on AGENTPACK, alongside comparisons with the base models and models trained on EditCoder dataset. On Python benchmarks, training on EditCoder dataset and AGENTPACK shows significant improvement over the base models for DeepSeekCoder and CodeLlama. The models trained on AGENTPACK typically outperform the EditCoder models, except for Qwen3-1.7B-Base. On JavaScript benchmarks, the models trained on AGENTPACK outperform their respective base versions. DeepSeekCoder-1.3B-Base (+0.04) and CodeLlama-1.3B-Base (+0.06) show substantial gains, while DeepSeekCoder-6.7B-Base (+0.01) and Qwen3-1.7B-Base (+0.02) exhibit marginal improvements.

**Ablations with Aider Edit Format** We perform an ablation study to compare our "ellipsis-hunk" edit format with the "diff" edit format used by Aider[1]. To assess the impact of the data format, we fine-tune the DeepSeekCoder models (1.3B and 6.7B) using a randomly sampled subset of our training data. We prepare two versions of this subset: one formatted with Aider's diff format and the other with our ellipsis-hunk format. We maintain the same training hyperparameters as our main experiments, but limit training to 1,000 steps to control computational costs. Additionally, we employ a specific prompt template (Figure 6) matching the Aider format for the evaluation of models trained with that style. Table 5 presents the results of this experiment. Our ellipsis-hunk format yields superior performance compared to the Aider format, showing an improvement of 0.07 and 0.16 in Pass@1 scores for the 1.3B and 6.7B models, respectively.

---

[1]https://aider.chat/docs/more/edit-formats.html#diff

# 6 LIMITATIONS

While AGENTPACK is large corpus of code edits co-authored by humans and software engineering agents, it has some limitations. (1) The dataset contains model-authored descriptions and corresponding code edits, but not the original prompts that elicited them. As a result, we do not observe the full interaction loop between programmers and agents. Nevertheless, the descriptions are often detailed, summarizing intent and rationale with sufficient clarity to serve as effective fine-tuning data. (2) The precise identity of the models that generated the edits is not always known. We expect that Claude Code commits originate from Anthropic models and Codex commits from OpenAI models, but Cursor supports a variety of back-end models, which makes attribution uncertain. (3) Agent-authored changes are produced in the context of an entire software repository, often involving additional files, project-specific configurations, and conversations that are not captured in the dataset. Consequently, some edits may rely on context that is not readily available. (4) Many changes in AGENTPACK are likely refined or further edited by human programmers before being merged. This is not strictly a limitation: AGENTPACK should *not* be interpreted solely as a measure of agent capability. Instead, it reflects the outcome of successful collaborations between humans and agents. It is likely that failed attempts to use an agent were never committed to GitHub. (5) The dataset is restricted to public repositories on GitHub and likely omits a significant amount of agent-authored code. For example, the Cursor Agent can be invoked in several ways, and only a few of them visibly sign their activity. The programmer can also suppress the signatures that we look for. Finally, a programmer could sign a commit "Co-authored by Claude" when they didn't use the agent at all, but this seems unlikely. (6) Our fine-tuning experiment showed that model can learn effectively from lightly filtered AGENTPACK. However we have not explored other filtering methods or other training setups as well as prompts with longer edit instructions. We do not investigate alternative filtering strategies, training formats, or prompt styles, and we do not isolate the impact of individual design choices. A more systematic exploration of these factors is an important direction for future work.

# 7 CONCLUSION

We introduced AGENTPACK, a large-scale corpus of 1.3M real-world code edits co-authored by software engineering agents and humans, curated from public GitHub activity between April and mid-August 2025. Unlike prior commit corpora, AGENTPACK contains long, LLM-written rationales as commit messages and many multi-file, non-trivial patches. These properties make it an effective dataset to study LLM agent behavior and model training. Fine-tuning code models on AGENTPACK yields consistent gains on two established editing benchmarks. We intend to continue expanding AGENTPACK to capture the ongoing activity of software engineering agents, and hope it will be a valuable resource for model development and understanding how programmers use agents in the wild.

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

# A APPENDIX

## A.1 FILE EXTENSION CLASSIFICATION

We classify files in AGENTPACK based on their file extensions and filenames using a direct mapping system. Each file is assigned to exactly one category based on its extension or, when no extension exists, its complete filename.

**Individual Programming Languages:**

- **Python:** `.py`, `.ipynb`
- **JavaScript:** `.js`, `.jsx`
- **TypeScript:** `.ts`, `.tsx`
- **Java:** `.java`
- **C:** `.c`, `.h`
- **C++:** `.cpp`, `.cc`, `.hpp`
- **C#:** `.cs`, `.csproj`
- **Go:** `.go`
- **Rust:** `.rs`
- **Swift:** `.swift`
- **Ruby:** `.rb`
- **PHP:** `.php`

- **Dart:** `.dart`
- **Kotlin:** `.kt`
- **Scala:** `.scala`
- **Shell:** `.sh`
- **Shell:** `.bat`
- **SQL:** `.sql`
- **Julia:** `.jl`
- **R:** `.r`
- **MATLAB:** `.m`
- **Lua:** `.lua`

**Web and Markup Languages:**

- **HTML:** `.html`
- **CSS:** `.css`
- **Markdown:** `.md`
- **Vue:** `.vue`
- **EJS:** `.ejs`
- **Svelte:** `.svelte`
- **Astro:** `.astro`

**DataFiles (Grouped Category):** `.json`, `.jsonl`, `.csv`, `.txt`, `.xml`, `.svg`, `.pdf`, `.png`, `.jpg`, `.jpeg`, `.gif`

**Config (Grouped Category):** `.toml`, `.yaml`, `.yml`, `.ini`, `.cfg`, `.conf`, `.env`, `.mk`, `.gradle`, `.properties`, `dockerfile`, `.example`, `.prettierrc`, `.gitignore`, `.gitattributes`, `.dockerignore`, `.gitmodules`, `.lock`, `.template`, `.version`, `.env_template`, `license`, `.gitkeep`

**Other Categories:** All extensions not explicitly mapped above

## A.2 PROMPT TEMPLATES

We place here the exact prompt templates used in our fine-tuning and evaluation: HumanEvalFix and CanItEdit (Fig. 4), and the one-shot CanItEdit variant (Fig. 5).

```
Fix bugs in {function_name}

Buggy Solution:

{function_signature}{buggy_solution}
{test}

Fixed Solution:

{function_signature}
```

```
## Code Before:
{old_code}

## Instruction:
{instruction}

## Code After:
```

(a) HumanEvalFix prompt template.      (b) CanItEdit prompt template.

Figure 4: Prompt templates used in our main fine-tuning evaluations: (a) HumanEvalFix and (b) CanItEdit.

```
## Code Before
def add(a, b):
    return a + b

## Instruction:
Add a "sub" function that subtracts two numbers. Also write docstrings for
both functions and change a,b to x,y.

## Code After
def add(x, y):
    """Adds two numbers."""
    return x + y

def sub(x, y):
    """Subtracts two numbers."""
    return x - y

## Code Before:
{old_code}

## Instruction:
{instruction}

## Code After:
```

Figure 5: CanItEdit 1-shot prompt template.

```
# Instruction
{instruction}

# Changes
File main.py
<<<<<<< SEARCH
{buggy_solution}
======
```

Figure 6: Aider's diff format prompt template.

