# OpenReview forum: "AgentPack: A Dataset of Code Changes, Co-Authored by Agents and Humans"
_ICLR.cc/2026/Conference — Submitted to ICLR 2026_

### Official Review · Reviewer_XjER · 2025-10-22

**Soundness:** 3
**Presentation:** 4
**Contribution:** 2
**Rating:** 4
**Confidence:** 4

**Summary:**

This paper introduces AGENTPACK, a large-scale dataset of 1.3M code edits co-authored by AI agents and humans. The core hypothesis is that this data, curated from public repositories and implicitly filtered by human maintainers, is superior to traditional commit-mined data. The paper includes a comprehensive analysis of the dataset's properties and an evaluation of DeepSeekCoder (1.3B, 6.7B) models, finding that fine-tuning on AGENTPACK yields generally positive results compared to the prior EditCoder dataset.

**Strengths:**

- Originality: The central idea of mining agent-human collaborative edits is novel and highly relevant. As agent use becomes ubiquitous, this data source will only grow.

- Quality & Scale: The resulting 59GB dataset is an order of magnitude larger and more complex than prior work (e.g., EditCoder). It also captures a wide variety of languages and task types, as shown in the analysis.

- Clarity: The paper is well-written. The data collection, filtering, and processing pipeline are all clearly articulated.

**Weaknesses:**

- Marginal Performance Gains: The performance gains over EditCoder are not uniformly strong. While significant on the 1.3B model, the improvement is minimal on the more capable 6.7B model (+6.7% on HumanEvalFix) and even shows a minor regression (-2.4% on CanItEdit). This suggests the dataset's primary value might be in bootstrapping weak models, while stronger models may not benefit as much.

- Outdated Model Evaluation: For a paper targeting a 2026 conference, the choice of base models is a significant weakness. The experiments use DeepSeekCoder V1 (released Nov 2023), while DeepSeekCoder V2 (June 2024) and DeepSeek V3 (Dec 2024) have been available for over a year. This makes it difficult to assess the dataset's relevance. The key open question is whether AGENTPACK provides a meaningful signal to SOTA models or if its benefits are limited to older, less capable models.

- Mismatch in Evaluation Complexity: The paper's evaluation is limited to relatively simple, single-function benchmarks (HumanEvalFix, CanItEdit). This is a missed opportunity IMO, as the AGENTPACK dataset itself contains complex, multi-file edits. Evaluating on more challenging benchmarks like SWE-bench would have been a far more convincing validation of the dataset's utility.

**Questions:**

1. Could the authors provide results on more complex, multi-file editing benchmarks like SWE-bench? This would better align the evaluation with the claimed complexity of the AGENTPACK data.

2. The paper frames the verbose, LLM-generated instructions as a positive. However, does this create a distribution mismatch and potentially hurt model learning? Could training on these hyper-detailed descriptions harm performance when the model is given a concise, or even vague, instruction from a real user? If the model is trained on these "solution-aware" descriptions, is it actually learning the difficult tasks of problem-solving, intent inference, and bug localization? Or is it just learning a simpler translation from a detailed commit description to code?

3. The training format ("ellipsis-hunk") seems mismatched from the evaluation format (full file output). Why was this format chosen over others more suitable for model generation, such as OpenAI's [diff format](https://cookbook.openai.com/examples/gpt4-1_prompting_guide#reference-implementation-apply_patchpy) or [Aider's search-replace format](https://aider.chat/docs/more/edit-formats.html)? Have other training representations been explored?

---

> ### Author Response · Authors · 2025-11-21
>
> We thank the reviewer for the detailed comments and address the concerns and questions below.
>
> **Marginal performance gains.**
>
> Our goal in the fine-tuning experiments is to show that a lightly filtered AgentPack can match the performance of EditCoder, which is trained on heavily filtered datasets; this demonstrates that AgentPack is a practical dataset for fine-tuning code-editing models. In addition, in an updated run with a slightly different model setup, the 6.7B model fine-tuned on AgentPack slightly exceeds the corresponding EditCoder baseline. In addition, we showed that fine-tuning on AgentPack can substantially improve model performance on HumanEval-JS (Javascript) by at least 0.02 (Qwen3-1.7B-Base) and at most 0.06 (CodeLLama-7B). We report these numbers in the revised version (Table 4).
>
> **Outdated model evaluation.**
>
> We chose Qwen3-1.7B as the more recent, open-weight models that can be fine-tuned within our compute budget, and we find that AgentPack also can improve it on JavaScript (see Table 4).
>
> **Mismatch in evaluation complexity.**
>
> While multi-file evaluations on benchmarks such as SWE-bench are important, they require substantial repository-level setup and engineering effort that is orthogonal to the dataset construction itself and beyond the scope of this dataset-introduction paper. We therefore focus on representative edit-level benchmarks and explicitly highlight large-scale, end-to-end evaluations on multi-file tasks as an important direction for future work.
>
> **Response to questions.**
>
> > Could the authors provide results on more complex, multi-file editing benchmarks like SWE-bench? This would better align the evaluation with the claimed complexity of the AGENTPACK data.
>
> See response to “Mismatch in evaluation complexity” above.
>
> > The paper frames the verbose, LLM-generated instructions as a positive. However, does this create a distribution mismatch and potentially hurt model learning? Could training on these hyper-detailed descriptions harm performance when the model is given a concise, or even vague, instruction from a real user? If the model is trained on these “solution-aware” descriptions, is it actually learning the difficult tasks of problem-solving, intent inference, and bug localization? Or is it just learning a simpler translation from a detailed commit description to code?
>
> We agree that commit messages are often more detailed than end-user instructions, but in our experiments we intentionally evaluate the model under concise benchmark prompts in HumanEvalFix and CanItEdit, not under the long descriptions in AgentPack. We do not observe performance degradation relative to EditCoder on these shorter prompts in most settings, suggesting that exposure to detailed rationales does not prevent the model from handling concise instructions. We clarify this point and note that studying robustness to varying prompt specificity is an important direction for future work in the updated draft.
>
> > The training format (“ellipsis-hunk”) seems mismatched from the evaluation format (full file output). Why was this format chosen over others more suitable for model generation, such as OpenAI’s diff format or Aider’s search-replace format? Have other training representations been explored?
>
> We chose the “ellipsis-hunk” representation because this format reduces sequence length, focuses the model on the change rather than the unchanged code, and allows training on large numbers of edits under the 4,096-token limit for DeepSeekCoder series of models.
>
> To address your suggestion regarding alternative representations, we have conducted a new ablation study comparing our “ellipsis-hunk” format against Aider’s diff edit format. We fine-tuned DeepSeekCoder models (1.3B and 6.7B) using both formats under controlled settings. Our results indicate that the “ellipsis-hunk” format yields superior performance, showing improvements of 0.07 and 0.16 in pass@1 scores for the 1.3B and 6.7B models, respectively. We have included these findings in a new paragraph of the revised paper, “Ablations with Aider Edit Format.”, in Section 5.

---

### Official Review · Reviewer_nPd5 · 2025-10-31

**Soundness:** 3
**Presentation:** 3
**Contribution:** 3
**Rating:** 4
**Confidence:** 4

**Summary:**

This paper introduces AGENTPACK, a large-scale dataset of 1.3M code edits (~60GB) co-authored by LLM-based coding agents (Claude Code, OpenAI Codex, Cursor Agent) and humans, mined from GitHub between April–Aug 2025. The authors argue that agent–human commits exhibit clearer natural language descriptions, more structured changes, and higher quality due to human filtering (merged PRs). They describe a pipeline to detect agent commits, extract diffs, filter dependency files, and align metadata. The paper analyzes code/description properties and task types, and demonstrates that fine-tuning models on AGENTPACK improves HumanEvalFix and CanItEdit performance over CommitPackFT and EditCoder datasets

**Strengths:**

Relevant and timely dataset capturing the rise of coding agents.
Solid motivation: agent-written commits include richer natural language intent than human-only commits.
Careful filtering: merged commits only; node_modules removal.
Extensive dataset analysis across languages, file types, patch sizes, and edit tasks.
Demonstrated performance gains over strong editing baselines (CommitPackFT, EditCoder).
Addresses an emerging paradigm shift in software engineering: human–LLM collaboration.
Useful resource for future research in code editing and agent behavior.

**Weaknesses:**

1. **Evaluation scope too limited**
   - Only Python subset used, despite claiming multi-language strengths.
   - Benchmarks are relatively small; improvements modest.
   - No test on real repositories or multi-file editing despite dataset emphasis.

2. **Assumptions not empirically validated**
   - “Merged commits = high quality”
   - “LLM commit messages = true intent”
   - No human study confirming commit message correctness or utility.

3. **Agent attribution uncertain**
   - Signature-matching may miss cases or misattribute commits.
   - Cursor backend ambiguity creates label noise.

4. **Missing critical ablations**
   - Quantity vs quality effect not disentangled.
   - No filtering strategy ablation.
   - No comparison to synthetic R1/agent-rationale datasets.

5. **Licensing and usage implications unclear**
   - Commercial training concerns left ambiguous.
   - No discussion of potential security or insecure-code contamination.

6. **Underexplored failure cases**
   - Where does AGENTPACK underperform?
   - Does training produce verbose or agent-like coding artifacts?

**Questions:**

1. How reliable is the agent detection pipeline? Did you manually sample for attribution accuracy?
2. Do LLM commit messages ever hallucinate rationale? Any validation?
3. Why not evaluate on JS/TS (largest portion of dataset)?

---

> ### Author Response · Authors · 2025-11-21
>
> Thank you for your insightful review. Here, we respond to the issues raised.
>
> **Evaluation scope too limited.**
>
> We provide the results of HumanEvalFix-JS (for JavaScript) in the updated version of the paper (Table 4 and Section 5). We focus on JavaScript due to existing HumanEvalFix support; extending to TypeScript is an interesting direction, but would require additional benchmark infrastructure.
>
> **Assumptions not empirically validated.**
>
> As explained to Reviewer vtF5, we have manually inspected a small random subset of AgentPack for quality assurance. This paper aims to present a curated dataset rather than evaluate downstream usage; our methodology treats commit messages as observational artifacts, consistent with established practice in Mining Software Repositories.
>
> **Agent attribution uncertain.**
>
> We have acknowledged possible omission of agent-authored code in our Limitation section (Point (5), Section 6). In addition, we have manually verified a random sample of commits per agent to make sure our signatures match the intended agent label. We made this clear in the updated paper. As for the backend ambiguity for Cursor Agent, we treat the agent identity (e.g., ‘Cursor Agent’) as the label of interest, so variation in Cursor’s backend model does not introduce label noise for our analysis..
>
> **Missing critical ablations.**
>
> We constructed AgentPack as a general dataset of agent activity rather than a controlled data-selection study, so we do not attempt to isolate causal effects of quantity or filtering choices. Our fine-tuning section is intended only as a feasibility demonstration that models can learn effectively from the curated corpus, not as an ablation-driven investigation of data quality. We will clarify this scope and soften any language that could be read as making stronger causal claims. Since R1/agent-rationale datasets target explicit reasoning traces, while AgentPack captures real-world edits where commit messages serve as developer-facing documentation rather than entire reasoning, our comparisons are limited to edit corpora like CommitPackFT and EditCoder.
>
> **Licensing and usage implications unclear.**
>
> Licensing and usage implications unclear: Similar to related projects (E.g. [GHArchive](https://github.com/igrigorik/gharchive.org)), we acknowledge that the commits may be bound by third party rights. We did not suggest commercial training as an intended use of AgentPack in the paper.
>
> **Underexplored failure cases.**
>
> Our fine-tuning results are intended only as a feasibility check, not as a comprehensive behavioral analysis. We clarify in the revised text that identifying failure modes or stylistic artifacts lies beyond the scope of this dataset paper and is an important direction for follow-up work.
>
> **Responses to questions.**
>
> > How reliable is the agent detection pipeline? Did you manually sample for attribution accuracy?
>
>  See response to “Agent attribution uncertain” above.
>
> > Do LLM commit messages ever hallucinate rationale? Any validation?
>
>  Our small quality-assurance check did not surface clear cases of fabricated or hallucinatory rationales. While this does not rule out errors, it indicates that such behavior is not prevalent within the sampled portion of the dataset, and we describe this check in the revised paper.
>
> > Why not evaluate on JS/TS (largest portion of dataset)?
>
>  See response to “Evaluation scope too limited” – we now provide results for the JavaScript subset of HumanEvalFix in the updated paper (Table 4 and Section 5).

---

### Official Review · Reviewer_vtF5 · 2025-10-31

**Soundness:** 3
**Presentation:** 3
**Contribution:** 3
**Rating:** 8
**Confidence:** 5

**Summary:**

This paper introduces AgentPack, a valuable dataset that captures code changes co-authored by LLM agents and human developers and successfully merged into production branches. The dataset’s methodology of using human acceptance as a critical quality filter, combined with the LLM’s ability to generate rich context, results in a corpus superior to historical human-only commit data for training advanced code-editing models. The contribution is significant and timely for advancing software engineering agent research.

**Strengths:**

1. Existing datasets for code changes are known to be noisy, with several tangled commits. In this work, the authors explicitly focus on commits created by coding agents with humans as gatekeepers, including only those that successfully merged into the main/master branch. As a result, the dataset benefits from an inherent human-in-the-loop filtering, ensuring that the data captures production-ready changes for training reliable code editing models.

2. The LLM-generated commit messages are not merely longer (median ~10x), but provide a qualitatively superior, explicit rationale for the change. This rich context is invaluable for teaching models the complex mapping between a detailed, single-purpose instruction (the "intent") and the corresponding code implementation.

3. The dataset captures structurally multi-hunk, multi-file code changes, often spanning large patch sizes (median 70 lines). The authors correctly point out that these are often logically unitary changes (e.g., implementing one feature across several architectural layers) that older datasets wrongly filtered out, providing a more realistic and challenging benchmark for agents.

**Weaknesses:**

1. The main limitation, IMO, is that the dataset is defined by the outcome (i.e., the merged commit) rather than the process (i.e., the full agent-human interaction history). As a result, it is not possible to track the agent trajectory, as well as the actual contribution of the human developers in refining it. This means the dataset trains models on a "successful collaborative output"rather than pure agent autonomy, limiting how much can be inferred about the agent’s standalone capability.

2. The claim that AgentPack commits are inherently "less tangled" compared to human-only data is hard to fully accept given their median size (70 lines across 2-3 files). A commit of this magnitude has a high structural possibility for introducing logical tangling, even if the agent's intent was singular. The paper’s reliance on the length and detail of the commit message to define quality and untangledness is a strong, yet potentially insufficient, assumption. A qualitative analysis categorizing the logical unity of a sample of these large commits would significantly strengthen this claim.

**Questions:**

1. What criteria or methodology were used to conclusively determine that a long, multi-file commit is logically focused rather than merely a combination of several documented changes?

2. Given the risk of LLM hallucination, how did the authors verify or ensure that these detailed rationales are factually correct and accurately describe the changes made to the code, rather than fabricating justifications? Is there a metric provided for the correctness of the commit messages?

---

> ### Author Response · Authors · 2025-11-21
>
> Thank you for your thoughtful review, specifically your acknowledgment of the realism and quality of AgentPack. Here, we address a number of your comments and questions.
>
> **Dataset defined by the outcome.**
>
> We are aware that AgentPack is defined by the outcome of human–agent collaboration, and we discuss this limitation under point (1) of the Limitations section (Section 6).
>
> **Logical tangling of commits and criteria for logical focus.**
>
> The intent of AgentPack is to contrast prior corpora that include many short commit messages with agent-coauthored commits whose descriptions more clearly articulate the intended change. However, we agree that a larger commit message, while providing more information, does not necessarily lead to a more focused commit. We have modified the wording in the paper to make this clearer and to avoid overstating any reduction in logical tangling.
>
> **Factual correctness of commit messages and rationales.**
>
> We have manually inspected a random sample of 10 commits per agent to verify that their commit messages describe the changes made to the code. There is no widely accepted automatic metric for factual correctness of commit messages or code-change rationales relative to the diff content. Prior work relies on either general natural-language metrics (e.g., BLEU/ROUGE/METEOR) or human annotation [1, 2]. The strongest automatic metric we found reaches only moderate discriminatory power (AUC = 0.75) on selected LLM/dataset combinations [3], meaning it can flag suspicious commit messages but cannot determine their factual correctness.
>
> [1] Dey, Samanta; Vinayakarao, Venkatesh; Gupta, Monika; Dechu, Sampath. 2022. Evaluating Commit Message Generation: To BLEU or Not To BLEU? In Proceedings of the 44th International Conference on Software Engineering — New Ideas and Emerging Results (ICSE ’22 NIER), May 10–11 2022, Pittsburgh, PA, USA. ACM, New York, NY. DOI:10.1145/3510455.3512790. [https://dl.acm.org/doi/10.1145/3510455.3512790](https://dl.acm.org/doi/10.1145/3510455.3512790)
>
> [2] Hu, Xing; Chen, Qiuyuan; Wang, Haoye; Xia, Xin; Lo, David; Zimmermann, Thomas. 2022. Correlating Automated and Human Evaluation of Code Documentation Generation Quality. ACM Transactions on Software Engineering and Methodology, Vol. 31, No. 4, Article 63 (July 2022). ACM, New York, NY. DOI:10.1145/3502853. [https://doi.org/10.1145/3502853](https://doi.org/10.1145/3502853)
>
> [3] Liu, Chunhua; Lin, Hong Yi; Thongtanunam, Patanamon. 2025. Hallucinations in Code Change to Natural Language Generation: Prevalence and Evaluation of Detection Metrics. arXiv preprint arXiv:2508.08661. DOI:10.48550/arXiv.2508.08661. [https://www.arxiv.org/pdf/2508.08661](https://www.arxiv.org/pdf/2508.08661)

---

### Official Review · Reviewer_7ETF · 2025-11-01

**Soundness:** 1
**Presentation:** 2
**Contribution:** 1
**Rating:** 2
**Confidence:** 3

**Summary:**

This paper introduces AgentPack, a corpus of 1.3M code edits co-authored by Claude Code, OpenAI Codex, and Cursor Agent across public GitHub projects. This corpus can be used to fine-tune large language models for code editing. Empirical results in this paper show that models fine-tuned on AgentPack can outperform models trained on prior human-only commit corpora.

**Strengths:**

1. This paper proposes a new dataset co-authored by agents and humans for fine-tuning LLMs on code editing tasks.
2. Empirical results show that models like DeepSeekCoder fine-tuned on the proposed corpus outperform models trained on prior human-only commit corpora.
3. This paper is easy to follow.

**Weaknesses:**

1. There are limited empirical results. For example, in Table 4, this paper only fine-tuned one LLM, i.e., DeepSeekCoder, and only compared with one prior training dataset EditCoder. More LLMs like CodeLlama and Mixtral can be included in the experiments, to make the results more solid.
2. Neither codebase nor dataset has been released to ensure reproducibility.
3. Some settings are unclear. Which exact model does Claude Code and Codex use?

**Questions:**

See Weaknesses.

---

> ### Author Response · Authors · 2025-11-21
>
> Thank you for your review. We would like to address the weaknesses you pointed out:
>
> **Limited model selection.**
>
>   We have updated our paper with results for fine-tuning CodeLlama-7B and Qwen3-1.7B (Results, Section 5).
>
> **No codebase or dataset released.**
>
>   We have released our dataset but the link is redacted to maintain anonymity. We will include the link to access the dataset in the camera-ready version.
>
> **Model setting for agents.**
>
>   Unfortunately, we do not know the proprietary models underlying Claude Code, Codex, or Cursor Agent. In the paper (Section 6), we explicitly state this limitation and clarify that our labels refer to the agent tool (e.g., “Cursor Agent”), not the underlying base model.

---

### Meta-Review · Area_Chair_bG5i · 2026-01-07

**Summary:**

This paper introduces AGENTPACK, a large-scale corpus of 1.3M real-world code edits co-authored by humans and software engineering agents, collected from public GitHub repositories. Compared to prior human-only commit datasets, AGENTPACK features richer natural-language rationales and more complex, multi-file edits. Analyses reveal adoption trends and distinctive structural properties of agent-assisted commits. Fine-tuning code models on AGENTPACK consistently improves performance on established code-editing benchmarks, demonstrating the value of agent-generated public data for training future code models.

**Reviewer Concerns:**

At the beginning, several concerns as follows: (1) Limited evaluation scope: Experiments fine-tune only one LLM (DeepSeekCoder V1) and compare against a single dataset, leaving generalization to stronger or newer models unclear. (2)Outdated baselines: The use of older models weakens the paper’s relevance for current state-of-the-art code LLMs. (3) Mismatch between data and evaluation: Benchmarks focus on small, single-function edits, despite AGENTPACK emphasizing complex, multi-file commits. (4) Dataset assumptions unverified: Claims about commit quality and untangled structure rely mainly on verbose commit messages, without human or qualitative validation. (5) Attribution and reproducibility issues: Agent detection may be noisy, and neither the dataset nor the code is released, limiting reproducibility.

After the rebuttal, I think several concerns still exist, such as a Limited evaluation scope and dataset quality.

**Reviewer Scores:**

The reviewers would keep their score unchanged.

---

### Decision · Program_Chairs · 2026-01-26

Reject